# Low Core Losses of Fe-Based Soft Magnetic Composites with an Zn-O-Si Insulating Layer Obtained by Coupling Synergistic Photodecomposition

**DOI:** 10.3390/ma15238660

**Published:** 2022-12-05

**Authors:** Siyuan Wang, Jingwu Zheng, Danni Zheng, Liang Qiao, Yao Ying, Yiping Tang, Wei Cai, Wangchang Li, Jing Yu, Juan Li, Shenglei Che

**Affiliations:** Research Center of Magnetic and Electronic Materials, College of Materials Science and Engineering, Zhejiang University of Technology, Hangzhou 310014, China

**Keywords:** Fe-based soft magnetic composite, photodecomposition, insulation layer, coupling treatment, ZnO

## Abstract

The major method used to reduce the magnetic loss of soft magnetic composites (SMCs) is to coat the magnetic powder with an insulating layer, but the permeability is usually sacrificed in the process. In order to achieve a better balance between low losses and high permeability, a novel photodecomposition method was used in this study to create a ZnO insulating layer. The effect of the concentration of diethyl zinc on the formation of a ZnO insulating film by photodecomposition was studied. The ZnO film was best formed with a diethyl zinc n-hexane solution at a concentration of around 0.40 mol/L. Combined with conventional coupling treatment processes, a thin and dense insulating layer was coated on the surface of iron powder in situ. Treating the iron powder before coating by photodecomposition led to a synergistic effect, significantly reduced core loss, and the effective permeability only decreased slightly. An iron-based soft magnetic composite with a loss value of 124 kW/m^3^ and an effective permeability of 107 was obtained at the frequency of 100 kHz and a magnetic field intensity of 20 mT.

## 1. Introduction

Soft magnetic composites (SMCs) allow the miniaturization and high frequency of electromagnetic devices, exhibit low eddy current losses over a relatively wide frequency range, and fill the application gaps between metal laminates and ferrites. Therefore, SMCs are promising candidates for the development of efficient and lightweight electromagnetic devices [1].

The coating of the insulating layer on a magnetic powder surface is the core process in the preparation of soft magnetic composites, and it can serve to suppress eddy currents in soft magnetic alloys. The materials used for insulation layers are often categorized into three categories: an organic insulation layer [2], an inorganic insulation layer [3,4,5,6,7,8], and an organic–inorganic composite insulating layer. The inorganic oxide materials commonly used to prepare coatings are mainly Al_2_O_3_ [3], ZrO_2_ [4], MgO [5], Fe_3_O_4_ [6], and ferrite [7]. There is also a novel Li-Al-O insulating layer [9].

For soft magnetic composites moving towards miniaturization and light weight, in the case of medium-frequency applications, it is necessary to reduce the magnetic loss so as to obtain high permeability. However, during the insulation coating process, it is difficult to achieve a balance between the effective permeability and the core loss, with low losses often decreasing the permeability.

For this reason, the following three requirements are proposed in the preparation of soft magnetic composites. First, the pressure needs to be as high as possible to reduce the presence of pores and to obtain a dense material. Second, the coating process needs to minimize the introduction of non-magnetic phases. Third, the insulating layer should be thin, dense, and uniform to the greatest extent possible, forming a relatively tight bond with the matrix phase so that the insulating layer can firmly adhere to the matrix phase.

A variety of methods for coating, such as the sol–gel method [10], the ball milling method [11], the bonding method [12], and the liquid phase method [13], have been used, all having their own advantages and disadvantages. But in most coating processes, it is very difficult to obtain an ideal coating, i.e., as thin as a mono-atom layer, fully dense, continuous and with a high electric resistance.

In this paper, a novel method with a controllable reaction process is proposed, which can coat a thin, dense, and uniform insulating layer on the surface of soft magnetic powder. It was expected that a sample with low loss and high permeability could be obtained using this method.

## 2. Experimental

### 2.1. Procedure

Reduced iron powders (purity > 99.5%; 0.2% Si, 0.003% P, and 0.015% S) provided by Jilin Huaxing Powder Metallurgy Technology Co. Ltd. with an average powder size of 75 μm were used as the matrix material.

The process of coating the insulating layer on the surface of the iron powders was as follows. Firstly, the silane coupling agent (KH550, 5 wt.% of the weight of the magnetic powder) was fully mixed with ethanol solution, then the iron powders were poured into it and stirred, and the surface of the iron powders was coated by ultrasonic treatment for 30 min. Secondly, after coupling treatment, the iron powders were washed with anhydrous ethanol many times and then transferred to a quartz glass bottle. Additionally, diethyl zinc [Zn(C_2_H_5_)_2_] solution and n-ethane (C_6_H_14_) solvents were added to the quartz glass bottle. Since diethyl zinc can easily be oxidized and hydrolyzed when encountering air, it was required to add diethyl zinc in a glove box with an oxygen content and water content of less than 5 ppm. Next, the quartz glass bottle was irradiated with an ultraviolet lamp with a wavelength range of 610 nm to 500 nm to achieve the photodecomposition of diethyl zinc (as shown in Figure 1). In the process of photocatalysis, a certain flow of air was regularly introduced into the quartz glass bottle. After a period of photodecomposition, the iron powders were removed by centrifugation and then transferred to a vacuum-drying oven and dried at 80 °C for 3 h.

The preparation process of the Fe-based soft magnetic composites was as follows. Zinc stearate (0.5 wt.%) acted as the lubricant and was mixed with the as-dried powders in the agate mortar. Then, the fully mixed powders were compacted at 1000 MPa into a ring with an outer diameter of 12.7 mm and an inner diameter of 7.5 mm. Finally, the rings were annealed at 773 K for 3 h under Ar atmosphere [14].

### 2.2. Characterization

The morphology of the coated Fe powders was observed via scanning electron microscopy (SEM, Hitachi SU1510, Tokyo, Japan). The element content of the coating layer was analyzed via X-ray fluorescence (XRF, ZSX PrimusII, Rigaku, Tokyo, Japan). The microscopic structures and elemental distribution of the polished cross-sectional Fe-based soft magnetic composite samples were analyzed via field-emission scanning electron microscopy (FESEM, Tescan mira3 Zeiss sigma 500, Brno, The Czech Republic) coupled with an energy dispersive spectrometer (EDS). Effective permeability was tested using an LCR meter (E4980A, Agilent, Santa Clara, CA, USA) with a frequency from around 5 kHz to 300 kHz at a voltage of 500 mV, based on Equation (1):(1)$$\mu_{e}=L\cdot l_{e}\cdot 10^{-2}/\left(0.4\pi \cdot A_{e}\cdot N^{2}\right)$$
where *L* is inductance; *l_e_* and *A_e_* are the effective length and area of the magnetic path, respectively; and *N* is the number of turns [15]. The core losses of the SMC rings were measured using a B–H curve analyzer (SY-8218, IWATSU, Tokyo, Japan, with less than 0.5% tolerance) in a frequency ambit of 20–100 kHz and with a magnetic excitation level of 20 mT.

The total core loss (*P_cv_*) of the sift magnetic materials can be divided into hysteresis loss (*P_h_*), eddy current loss (*P_ed_*), and residual loss (*P_exc_*) according to the generation mechanism; the relationship can be expressed by the following Equation (2):(2)$$P_{cv}=P_{h}+P_{ed}+P_{exc}=K_{H}B_{m}^{\alpha }f+\frac{K_{E}D^{2}B_{m}^{2}f^{2}}{\rho }+P_{exc}$$
where *K_H_* and *K_E_* are constants; *B_m_* is magnetic flux density (mT); ρ is the resistivity (Ω⋅m); *D* is the grain size, which can be regarded as a constant for a certain sample; α corresponds to the Steinmetz coefficient; and *f* is the frequency [16]. Usually, α = 1–2 for SMCs and α = 2–3 for ferrite materials.

When *P_cv_* is divided by *f*, Equation (2) can be expressed as Equation (3):(3)$$\frac{P_{cv}}{f}=\frac{P_{h}}{f}+\frac{P_{ed}}{f}+\frac{P_{exc}}{f}=K_{H}B_{m}^{\alpha }+\frac{K_{E}D^{2}B_{m}^{2}f}{\rho }+\frac{P_{exc}}{f}$$

By measuring the loss at different frequencies, a curve of $\frac{P_{cv}}{f}$ ~ *f* can be drawn. The intercept of the ordinate is the hysteresis loss ($\frac{P_{h}}{f}$). Then, the eddy current loss ($\frac{P_{ed}}{f}$) is separated, which is linearly related to the frequency. The rest of the non-linear part is related to the residual loss ($\frac{P_{exc}}{f}$).

## 3. Results and Discussion

Effect of Surface Treatment Methods on the Morphology and Magnetic Properties of Coated Iron Powders

The quality of the coating on the surface of iron powder depends on the compactness of the coating layer and the adhesion with the iron matrix. For this reason, the effects of different surface treatment methods on the formation of coating layer on the iron powder surface were studied, and the different morphologies are shown in Figure 2. Figure 2a shows the powder coated by ultraviolet light decomposition in the diethyl zinc solution. It can be observed that the surface of the iron powder had many products generated by photodecomposition, and the gullies in the iron powder were obviously filled with coarser particles.

Figure 2b shows the powder coated by ultraviolet light decomposition in the same mass diethyl zinc solution diluted to 0.4 mol/L by an n-hexane solvent. The surface insulation products appeared to be slightly fewer compared with those shown in Figure 2a, and the surface was smoother.

Figure 2c shows the powder coated by coupling with 5 wt.% KH550, and it can be seen that there was no significant change on the surface of the iron powder. This is due to the thinness of the organosilane film.

Figure 2d shows the powder coated by coupled with 5 wt.% KH550, and then coated by ultraviolet light decomposition in a mixed solution of 0.4 mol/L diethyl zinc and n-ethane. It can be seen that the coating layer on the powder surface was filmy, and some of the powder surface was scattered with some non-filmed insulating particles.

The iron powder samples obtained from the different solution coating treatments, shown in Figure 2, were analyzed by XRF, and the results are shown in Table 1. Although the accuracy of XRF is not very high, it can be seen that the sample powder treated with diethyl zinc solution only had a higher content of Zn or ZnO in the XRF elements, but as can be seen in Figure 2, the size of the particles produced by photodecomposition was large. However, more non-magnetic phase Zn or ZnO content will lead to the decrease of saturation magnetization (Ms) of magnetic particles (see Appendix A).

With respect to diethyl zinc with the same percentage content of iron powder, when the n-hexane solvent was added for dilution, the concentration of diethyl zinc decreased, and the corresponding proportion of zinc or zinc oxide obtained after photodecomposition was also reduced by about one-tenth.

When the iron powder surface was pretreated by coupling with 5 wt.% KH550, the Zn content was elevated relative to that without pretreatment, which indicated that the coupling treatment was beneficial to the formation of a zinc coating layer on the iron powder surface through photocatalytic decomposition. This may be because the coupling modification of the surface by KH550 reduced the agglomeration between the surface particles and increased the specific surface area of the powder, which resulted in more Zn or ZnO deposited on the surface of the sample [17].

Figure 3 shows the core loss and effective permeability of the magnetic ring pressed with iron powders coated using different treatments methods. It can be seen that the highest core loss and the poor high-frequency stability of the permeability were obtained when the iron powders were coated using photolysis with only the diethyl zinc solution. When the iron powders were coated with the product of the photodecomposition of the diethyl zinc diluted with n-hexane solvent, the core loss of the sample was reduced by about two-thirds, and the high-frequency stability of the permeability was better compared to that of the samples prepared without the n-hexane solvent. When the iron powder was pre-coupled with KH550 and then coated with a low concentration of diethyl zinc solution, it was found that the core loss was significantly reduced and the effective permeability not only remained at a high value of about 107, but also had excellent high-frequency stability. Coupling treatment and photodecomposition coating were integrated together and produced good results, which cannot be achieved by using these treatments individually. In order to further explore the influence of different surface treatment methods on the core loss, the measured core loss values of the three samples were separated, respectively.

We referred to the formula according to the Bertotti’s loss separation model [18], *P_cv_* = *P_ed_* + *P_h_*
_+_
*P_exc_*, where *P_cv_, P_ed_*_,_
*P_h_*, and *P_exc_* represent total core loss, eddy current loss, hysteresis loss, and residual loss, respectively. Hysteresis loss (*P_h_*) is generally related to the powder particle size, internal stress, and the purity of the entire matrix phase. Eddy current loss (*P_ed_*) is related to material resistivity and thickness. Residual loss (*P_exc_*) is closely related to the magnetization rate [19]. When the frequency is below 500 KHz, residual loss (*P_exc_*) is usually very small for iron-based SMCs; most of the core loss is from hysteresis loss (*P_h_*) and eddy current loss (*P_ed_*) [20,21].

The loss separation results for the iron-based soft magnetic composites prepared by different surface treatments are shown in Figure 4 (at *B* = 20 mT, *f* = 100 kHz). As previously discussed, when diethyl zinc was not diluted, the ZnO content produced by photodecomposition was higher. For magnetic particles, the introduction of ZnO increased the impurity mass. Therefore, the magnetic hysteresis loss of the sample coated by photodegradation of undiluted diethyl zinc was higher than that diluted with ethane.

The integrity and uniformity of the insulating layer on the surface of iron powder are conducive to improving the resistivity of the composite magnetic ring, thus reducing eddy current loss. The coarse ZnO grains generated by photodecomposition in a high concentration of diethyl zinc solution led to the formation of a thick and discontinuous insulation layer. However, after the concentration of diethyl zinc was diluted, the ZnO grains generated were fine. In addition, after the organic silane film was obtained using the pre-coupling treatment for the iron powder, it was coated with ZnO. The silane film and ZnO played a synergistic role, which increased the adhesion between the coating and the substrate and formed a uniform insulating layer, thereby greatly reducing eddy current loss.

To confirm the formation of a homogeneous insulating layer on the surface of the sample as predicted by the loss separation data, the sample treated using pre-coupling and subsequent photodecomposition was polished on the cross-section, and the element distribution around the iron particles was observed by EDS, as shown in Figure 5. It can be seen from the figure that the iron particles were isolated by Zn, O, and Si elements, and the distribution of these three elements was consistent in the area. This proves that a relatively uniform and dense Zn-O-Si insulation layer was formed. It can also be seen that the thickness of the insulating layer was small, which verifies that the photodecomposition method can indeed deposit a thin, dense, and uniform insulating layer. The thin insulating layer caused a weaker magnetic dilution effect and was able to obtain a higher magnetic permeability. This was also consistent with the high effective permeability shown in Figure 3b.
(4)Surface−OH+H2N(CH2)3Si(OCH)3+H2O→Surface(O−Si)nOH+CH3OH
(5)Surface(O−Si)nOH+C2H5−Zn−C2H5→Surface(O−Si)nO−Zn−C2H5+C2H6
(6)Surface(O−Si)nO−Zn−C2H5+7O2→hvSurface(O−Si)nO−Zn−O+C2H6

Combining the above results, the simultaneous use of coupling treatment and photodecomposition coating achieved remarkable performance improvements and a good synergistic effect. For this synergistic mechanism, Figure 6 shows our speculations regarding this. During the pre-coupling treatment, the hydroxyl group adsorbed on the surface of iron powder reacted with KH550 for dehydration, producing (O−Si)nOH organosilane film (see Equation (4)), whereas, after adding diethyl zinc, the diethyl zinc reacted with the hydroxyl group of organosilane film, bonding them to form (O−Si)nO−Zn−C2H5 (see Equation (5)) [22]. Under sufficient UV energy and an aerobic environment, the intermediate chemical bond of $-\mathrm{Zn}-C_{2}H_{5}$ is dissociated and generates ZnO (as in Equation (6)), which is deposited on the surface of iron powder together with organosilane film, thereby achieving a thin and dense insulating layer. IR of iron powder with different treatments are shown in Appendix A. After pre-coupling treatment and photodecomposition of iron powder, the characteristic absorption peaks of Si-O bond and Zn-O bond appear at 1077.58 cm^−1^ and 472.25 cm^−1^ respectively, which may prove the occurrence of Equation (6).

In order to further investigate the effect of the diethyl zinc concentration on the coating quality of the iron powder surface, we specified the premise of diethyl zinc relative to the mass of the magnetic powder, and diluted the diethyl zinc with an n-ethane solvent to obtain different molar concentrations of diethyl zinc solutions; additionally, photodegradation coating treatment was performed on the iron powder pre-coupled with 5 wt.% KH550. The morphology of the iron powder obtained is shown in Figure 7.

When the concentration of diethyl zinc was low, a cracked orange-peel-shaped insulating layer was formed on the surface of the iron powder, which is shown in the circle of Figure 7a. When the diethyl zinc concentration increased, the cracks on the surface insulation layer gradually started to become smaller, as shown in the circle in Figure 7b. When the concentration rose further to 0.40 mol/L, most of the surface insulating layer had started to form a film. Finally, when the concentration rose to 0.48 mol/L (as shown in Figure 7d), the orange-peel insulating layer appeared again, and the cracks of the surface insulating layer became larger, as shown in the circles.

Figure 8 shows the core loss and effective permeability of the magnetic ring pressed with iron powders coated by photolysis in diethyl zinc solutions of different concentrations. It can be seen that the concentration of the diethyl zinc solution had a large effect on the core loss, and the core loss value was first reduced but then increased again with the increase in the diethyl zinc solution concentration.

From this, we were able to determine the optimal solution concentration. Combined with the SEM images shown in Figure 7, it was found that the optimal concentration was 0.40 mol/L. At this time, the powder surface was essentially coated with a film similar to a Zn-O-Si compound. When the concentration of the solution was further increased, the core loss was not reduced but was, instead, increased, which corresponds to the rule of insulating film formation quality. In general, the effective permeability decreased with increasing diethyl zinc solution concentrations due to the magnetic dilution effect, but, in this section, the effective permeability first decreased and then increased.

We speculate that the photodecomposition products tended to be produced on the sample surface first, and when the diethyl zinc solution concentration was higher, they tended to be produced in the solution. Thus, the magnetic dilution effect was diminished again, and the effective permeability became higher. When the diethyl zinc solution concentration was 0.40 mol/L, the effective permeability was larger and more stable with regard to the frequency changes. However, when the concentration of KH550 was increased from 2 wt.% to 6 wt.%, the fluctuation of loss and effective permeability of the SMC rings was not as obvious as that of n-ethane, the relatively optimized KH550 concentration was 5 wt.% (see Appendix A).

In summary, after the pre-coupling treatment with 5 wt.% KH550, the iron powder surface was coated with an excellent insulating film by photolysis in the 0.40 mol/L diethyl zinc solution. The composite magnet prepared with the coated iron powders had an excellent core loss of 124 kW/m^3^ and an effective permeability of 107 when the magnetic field strength was 20 mT and the frequency was 100 kHz.

## 4. Conclusions

In this paper, a thin and dense insulating layer was coated on the surface of iron powder in situ using pre-coupling treatment and the photodecomposition of diethyl zinc so as to obtain samples with low core losses and high effective permeability. The concentration of diethyl zinc was found to be related to the size of the ZnO grains generated by photodecomposition, which also determined the denseness of the insulating film. A high concentration of diethyl zinc resulted in coarse ZnO grains, and the surface insulation products showed the appearance of cracked orange peel, while dilution by the n-hexane solvent resulted in fine zinc oxide grains. The iron powder was pre-coupled and then coated by photodecomposition, and the superposition of the two treatments achieved a synergistic effect, generating a composite insulation layer that was similar to Zn-O-Si. The insulation layer not only improved the adhesion to the matrix, but it was also more dense. The eddy current loss was obviously reduced, and the thin insulation layer weakened the magnetic dilution effect, so the effective permeability was high. At a magnetic field strength of 20 mT and a frequency of 100 kHz, the loss value was about 124 kW/m^3^ and the effective permeability was about 107.

## Figures and Tables

**Figure 1 materials-15-08660-f001:**
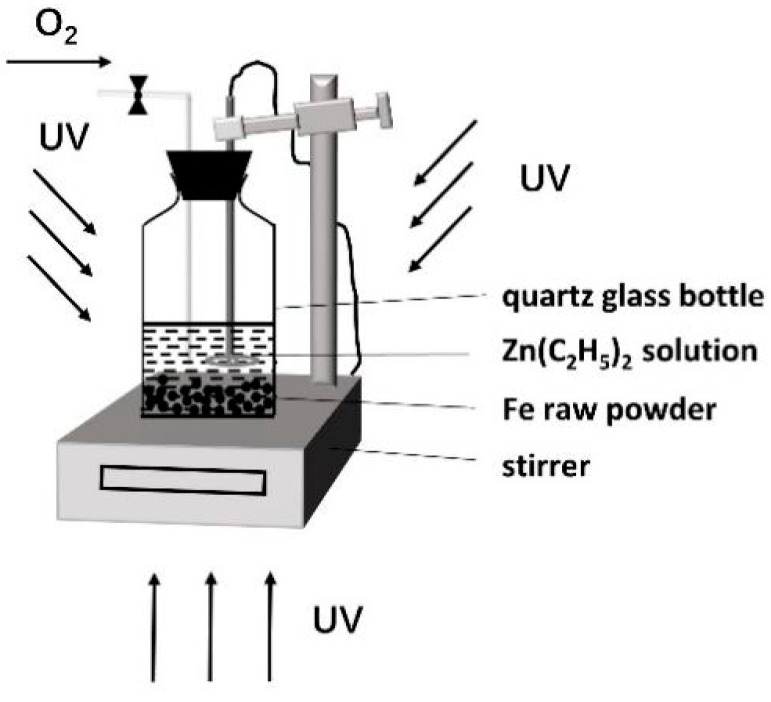
Schematic diagram of iron powders coated with ZnO by photodecomposition.

**Figure 2 materials-15-08660-f002:**
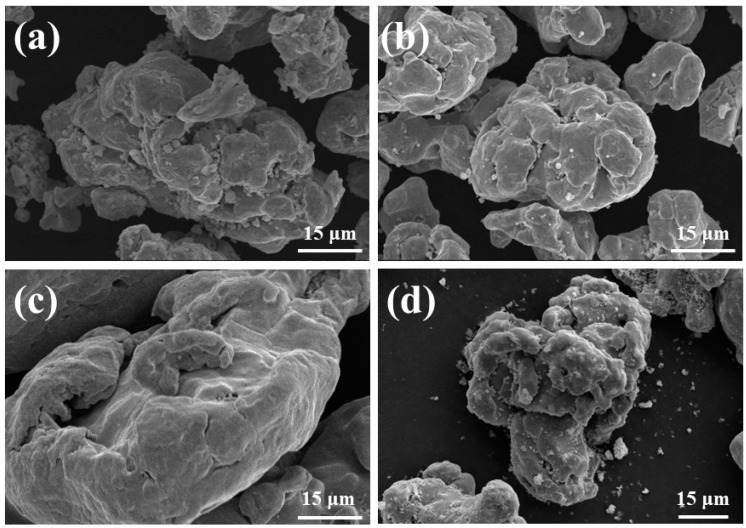
SEM of photolytically coated iron powder with different treatments: (**a**) Zn(C_2_H_5_)_2_, (**b**) Zn(C_2_H_5_)_2_ + C_6_H_14_, (**c**) 5 wt.% KH550, and (**d**) 5 wt.% KH550 and (Zn(C_2_H_5_)_2_ + C_6_H_14_).

**Figure 3 materials-15-08660-f003:**
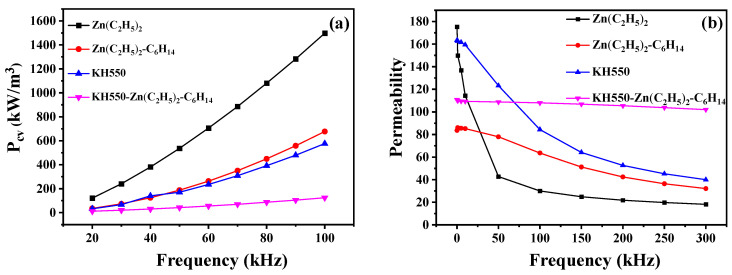
(**a**) Core loss and (**b**) effective permeability of the coated Fe-based magnetic ring prepared with different surface treatments.

**Figure 4 materials-15-08660-f004:**
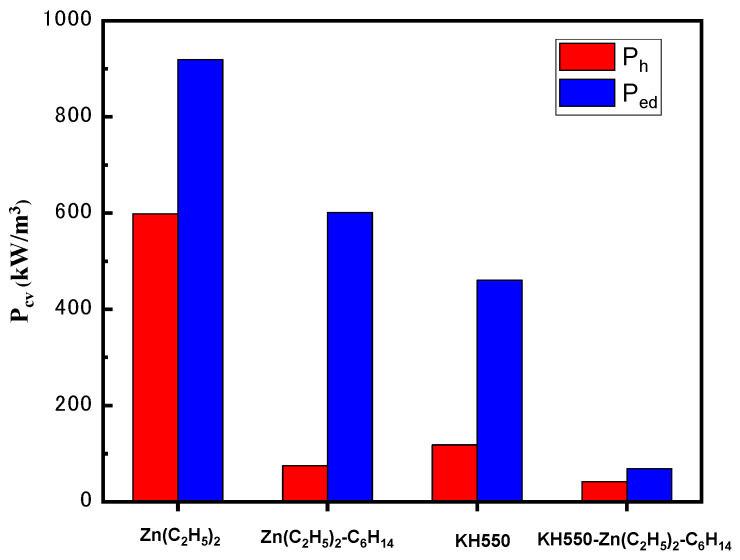
Separation of iron loss for samples prepared by different surface treatments at *f* = 100 kHz and *B* = 20 mT.

**Figure 5 materials-15-08660-f005:**
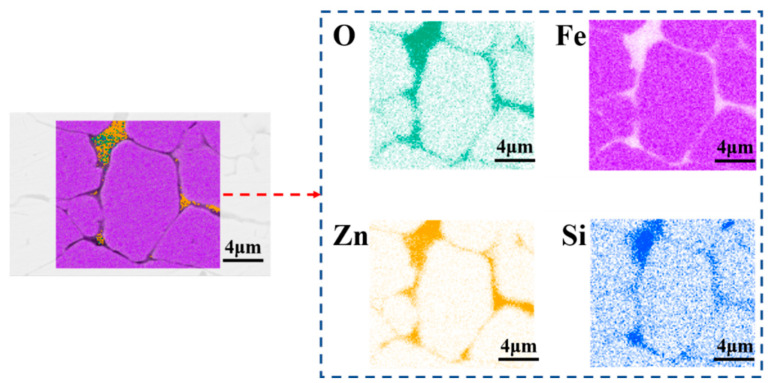
EDS spectra of polished cross-section samples coated with 5 wt.% KH550 + Zn(C_2_H_5_)_2_ + C_6_H_14_.

**Figure 6 materials-15-08660-f006:**
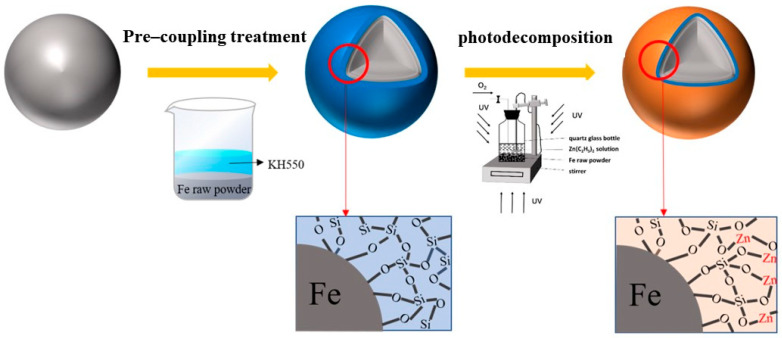
Schematic diagram of Fe@Zn-O-Si prepared using the pre-coupling and photodecomposition treatment.

**Figure 7 materials-15-08660-f007:**
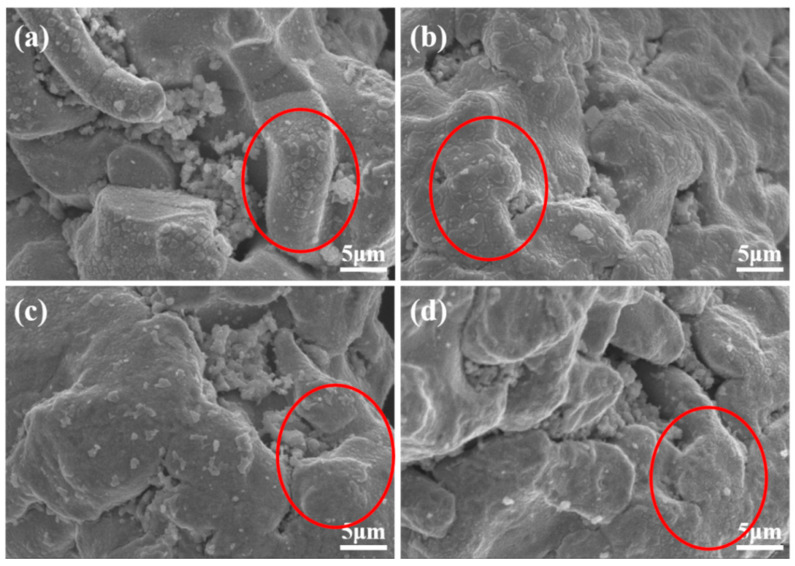
SEM images of iron powder coated using photolysis in diethyl zinc solutions of different concentrations: (**a**) 0.24 mol/L, (**b**) 0.32 mol/L, (**c**) 0.40 mol/L, and (**d**) 0.48 mol/L.

**Figure 8 materials-15-08660-f008:**
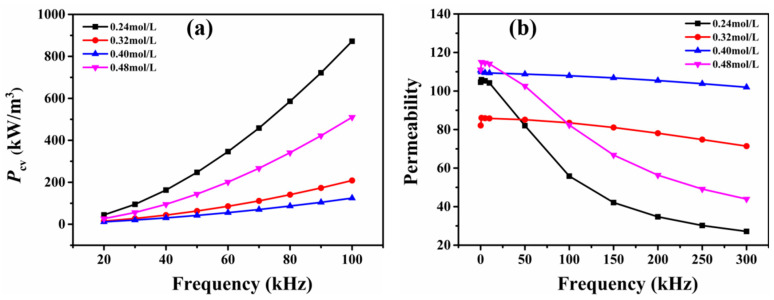
(**a**) Loss and (**b**) effective permeability of SMC rings made by the photodecomposition of coated iron powder with diethyl zinc solutions of different concentrations.

**Table 1 materials-15-08660-t001:** Elemental effects of different treatments on the prepared powders.

Element (at%)	Zn(C_2_H_5_)_2_	Zn(C_2_H_5_)_2_ + C_6_H_14_	KH550	KH550 and (Zn(C_2_H_5_)_2_ + C_6_H_14_)
Fe	88.27	98.26	99.65	97.57
Zn	11.73	1.74	0	2.17
Si	0	0	0.35	0.26

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
