# Peer review of "Low Core Losses of Fe-Based Soft Magnetic Composites with an Zn-O-Si Insulating Layer Obtained by Coupling Synergistic Photodecomposition"

_materials, 2022, doi:10.3390/ma15238660_

Round 1

Reviewer 1 Report

Comments to the Author

The authors prepared soft magnetic composites (SMC) by using iron powder zinc oxide and silicate. It showed quite stable structure and high magnetic property. The authors presented interest work but still remain some problems in this research. So, I would suggest an acceptance for publication after revision which is suggested as follows:

1. The iron structure was coated by ZnO and SiO2 and SEM image and EDS data were shown. But it is not enough to prove the structure. So, please show the FT-IR data to confirm the vibration of metal-oxygen bonding and XRD data for the analysis of lattice structure, or provide the XPS to measure the chemical binding energy.

2. The author fabricated the soft magnet structure which was composed with Zn-O-Si and claimed that this structure showed good properties. However, there is no control test. So, please compare the magnetic property of Zn-O-Si based soft magnet with only Zn-O based soft magnet to prove this structure showed better performance. Also if Si might play important role, please show the magnetic property depending on the amount of Si and optimize the condition.

3. In this study, it seems that the thickness of grain boundary that is insulating layer is important factor. So, is it possible to control the size of grain boundary and the property could be tuned by control of boundary size? If yes, please show the magnetic property of structure depending on the boundary size and optimize the condition.

Reviewer 2 Report

Review for materials-2011052

„Low core losses Fe-based soft magnetic composites with Zn-O-Si insulating layer obtained by coupling synergistic photodegradation” 

by Siyuan Wang , Jingwu Zheng  , Danni Zheng , Liang Qiao , Yao Ying , Yiping Tang , Wei Cai , Wangchang Li , Jing Yu , Juan Li , Shenglei Che

The authors report a novel photocatalytic decomposition method to generate an insulating ZnO layer on the surface of iron powder. This method is combined with a conventional coupling process to produce a thin and dense insulating layer deposited in situ on the surface of iron powder. The result is a homogeneous composite insulating layer similar to Zn-O-Si that significantly reduces magnetic loss and exhibits only a small reduction in effective magnetic permeability.

In principle, the work presented here is interesting and new enough to publish in Materials. Unfortunately, the presentation of the work and here especially the text is of very low quality.  Throughout the manuscript there is no clear definition of what the authors mean by magnetic loss (I assume they mean magnetic induction due to eddy currents). Here, a clear definition would help readers not deeply involved in this particular field of nanomagnetism to understand what the work done here is about. There are also unclear definitions and meanings in other parts of the text. Some references for example for the Bertotti formula are missing as well.

Another very disturbing point is that apparently no one checked spelling and grammar before submitting. The sentences are very long (sometimes more than 10 lines) and lack correct punctuation, making them unreadable. Equations 2-4 appear somewhere in the text and it is completely unclear where they belong.

In contrast, the figures are well presented, and if the text were of similar quality, this would clearly contribute to the acceptance of the paper. I therefore recommend a comprehensive language and grammar check and clarification of the above points before resubmitting the work.

Reviewer 3 Report

This work has some interest for specialists in the field, but there are some issues to be addressed. How many times were these measurements repeated? Pressing and sintering can be suspect to repeatability. The authors did not define Pexc nor did they reference Bertotti. Additionally, they did not explain how they determined the contribution of each mechanism to the total power loss. Finally, the EDS analysis is somewhat suspect. Since the different processes lead to coatings on the particles, the EDS analysis should be done with that in mind. Under normal conditions, EDS analysis assumes uniform elemental dispersion throughout the sample, which does not hold in the present case. The layering leads to an overestimation of the surface.

Reviewer 4 Report

The paper presents very interesting and probably useful results on novel SMCs, but the main focus is put on the preparation and structural characterization, while the analysis of magnetic properties is poor.

As SMCs are "magnetic" materials, the magnetic properties sholud be presented properly:

How was the power loss separation performed? Which equations and parameters were used?

Why are the excess loss components so low? - This could be true for e.g. steels, but for composites Pexc are expected to be higher...

Explain the magnetic characteristics in terms of magnetization processes.

Round 2

Reviewer 1 Report

Thanks for authors' effort to revise the manuscript, and I agree to accept this article for publication in Materials.

Author Response

Thank you very much for your kind review and comments.

Reviewer 3 Report

The authors need to clarify that XRF results do not represent the actual composition and in a composite material, the average bulk resistance is not the primary factor in the eddy current losses but depend on the particle size and the skin depth. Finally, for readability, in lines 145-150, the authors repeatedly redefine the variables for the various power losses which were first defined in lines 82-83. Using the symbols instead of symbols and words would help the reader.

Reviewer 4 Report

There are still some problems regarding the magnetic properties analysis:

- Eqs. 1-3 are without any reference.

 - For which conditions is Eq. 2 valid? (Why is Bm a power of 3 in Ph, when usually is 3/2...)

- The authors claim that Pexc are negligibly small because in ref. 17 it was so, but it is a rare case and in majority of SMCs Pexc are quite high, otherwise it must be explained in terms of magnetization processes. The authors should find more works regarding loss separation in various SMCs under various conditions...
